# Comparison of climatic factors on mosquito abundance at US Army Garrison Humphreys, Republic of Korea

Myung-Jae Hwang[1]☯, Heung-Chul Kim[2]☯, Terry A. Klein[2], Sung-Tae Chong[2], Kisung Sim[3], Yeonseung Chung[3], Hae-Kwan Cheong[1]*

**1** Department of Social and Preventive Medicine, Sungkyunkwan University School of Medicine, Suwon, Gyeonggi-do, Republic of Korea, **2** Force Health Protection and Preventive Medicine, Medical Department Activity-Korea/65th Medical Brigade, Unit 15281, United States of America, **3** Department of Mathematical Sciences, Korea Advanced Institute of Science and Technology, Daejeon, Republic of Korea

☯ These authors contributed equally to this work.
* hkcheong@skku.edu

**Data Availability Statement:** Data is available in VectorMap, a DoD repository for mosquito collection and pathogen detection housed at the

## Abstract

### Introduction

A number of studies have been conducted on the relationship between the distribution of mosquito abundance and meteorological variables. However, few studies have specifically provided specific ranges of temperatures for estimating the maximum abundance of mosquitoes as an empirical basis for climatic dynamics for estimating mosquito-borne infectious disease risks.

### Methods

Adult mosquitoes were collected for three consecutive nights/week using Mosquito Magnet® Independence® model traps during 2018 and 2019 at US Army Garrison (USAG) Humphreys, Pyeongtaek, Gyeonggi Province, Republic of Korea (ROK). An estimate of daily mean temperatures (provided by the Korea Meteorological Administration) were distributed at the maximum abundance for selected species of mosquitoes using daily mosquito collection data after controlling for mosquito ecological cycles and environmental factors.

### Results

Using the Monte-Carlo simulation, the overall mosquito population abundance peaked at 22.7°C (2.5th—97.5th: 21.7°C–23.8°C). *Aedes albopictus*, vector of Zika, chikungunya, dengue fever and other viruses, abundance peaked at 24.6°C (2.5th–97.5th, 22.3°C–25.6°C), while Japanese encephalitis virus (JEV) vectors, e.g., *Culex tritaeniorhynchus* and *Culex pipiens*, peaked at 24.3°C (2.5th–97.5th: 21.9°C–26.3°C) and 22.6°C (2.5th–97.5th: 21.9°C–25.2°C), respectively. Members of the *Anopheles* Hyrcanus Group, some of which are vivax malaria vectors in the ROK, abundance peaked at 22.4°C (2.5th–97.5th: 21.5°C–23.8°C).

### Conclusion

The empirical mean temperature ranges for maximum abundance were determined for each mosquito species collected at USAG Humphreys. These data contributed to the

Walter Reed Biosystematics Unit at: <http://vectormap.si.edu/>.

**Funding:** This research was supported by the Armed Forces Health Surveillance Branch, Global Emerging Infections Surveillance and Response System (AFHSB-GEIS), Silver Spring, MD (ProMIS ID #P0025-2018-ME). The opinions or assertions contained herein are the views of the authors, and are not to be construed as official, or as reflecting the views of the Department of the Army, or the Department of Defense. Authors, as employees of the U.S. Government (HCK, MSK, TAK), conducted the work as part of their official duties. Title 17 U.S.C. §105 provides that 'Copyright protection under this title is not available for any work of the United States Government'. Title 17 U.S.C. §101 defines a U.S. Government work is a work prepared by an employee of the U.S. Government as part of the person's official duties. The funders had no role in study design, data collection and analysis, decision to publish, or preparation of the manuscript.

**Competing interests:** The authors have declared that no competing interests exist.

**Abbreviations:** CDC, child development care; CIs, confidence intervals; $CO_2$, carbon dioxide; df, degree of freedom; DMZ, demilitarized zone; DPW, department of public works; GAM, generalized additive model; GLM, generalized linear model; KCDC, Korea Centers for Disease Control and Prevention; QAIC, quasi-Akaike Information Criterion; ROK, Republic of Korea; RR, relative risk; ULV, ultra-low-volume; USAG, US Army Garrison; TI, trap index; VBD, vector-borne disease.

identification of relative mosquito abundance patterns for estimating mosquito-borne disease risks and developing and implementing disease prevention practices.

## Introduction

Global climate change is expected to have a profound impact on human health due to the increasing impact of a multitude of vector-borne diseases (VBD) [1]. VBD, e.g., mosquito-borne infections, are closely related to climatic factors, e.g., temperature and precipitation [2]. Optimum temperatures not only affect the immature development, oviposition cycle, adult longevity, and contact with hosts, but also the extrinsic development of pathogens of medical and veterinary importance. Additionally, larval development is dependent upon water sources for species-specific habitats, e.g., tree holes/artificial containers, ponds, and stream margins, and associated aquatic vegetation. Together, these variables directly affect ecological mechanisms that negatively or positively impact relative mosquito population abundance, transmission of pathogenic agents, and potential for extended geographic distribution [3–5].

The Intergovernmental Panel on Climate Change predicted that during the 21st century that the average worldwide temperatures would continue to increase by 2.4˚C to 6.4˚C above previously recorded temperatures [6]. As a result, these increased temperatures and changes in precipitation patterns, as well as other meteorological factors, would affect ecosystems worldwide. These changes have a direct impact on seasonal and geographical distributions of mosquitoes and other arthropods that impact on human health [7–10]. Mosquito population abundance rapidly increases when environments are suitable for increased survival and rapid larval development. Additionally, the extrinsic pathogen development is decreased as temperatures increase to optimum levels, and in combination with higher population densities, increase the probability for the transmission of infectious agents of medical and veterinary health [11].

A mosquito-borne infectious disease surveillance system based on the worldwide mosquito monitoring system is being actively implemented. The objectives of the monitoring system are to estimate the relative abundance of mosquitoes over a time-based collection time-series data to inform public health officials to implement disease risk reduction strategies among military and civilian populations. Previous investigations have mostly observed the correlation between the relative abundance of mosquito populations and annual or monthly mean temperatures and precipitation [12, 13]. These estimates do not reflect the time-series changes of relative mosquito abundance based on seasonality and mosquito species life cycles. Mosquito growth and development occurs only within a limited range of temperatures, in addition to the requirement for associated breeding sites. Depending on the mosquito species and stage of development, the lower temperature limits are between 5˚C-10˚C, while the upper-temperature limits are between 30˚C-35˚C. Biologically unsuitable temperatures are detrimental for growth, resulting in a decreased population growth, and especially if adverse environmental conditions persist for a long time.

This study aimed to identify time-series patterns for species-specific mosquito abundances based on daily collection data during the 2018 and 2019 mosquito season at US Army Garrison (USAG) Humphreys, Pyeongtaek, Gyeonggi Province, Republic of Korea (ROK). Empirical temperature ranges for maximum mosquito abundance was determined for estimating mosquito-borne disease risks.

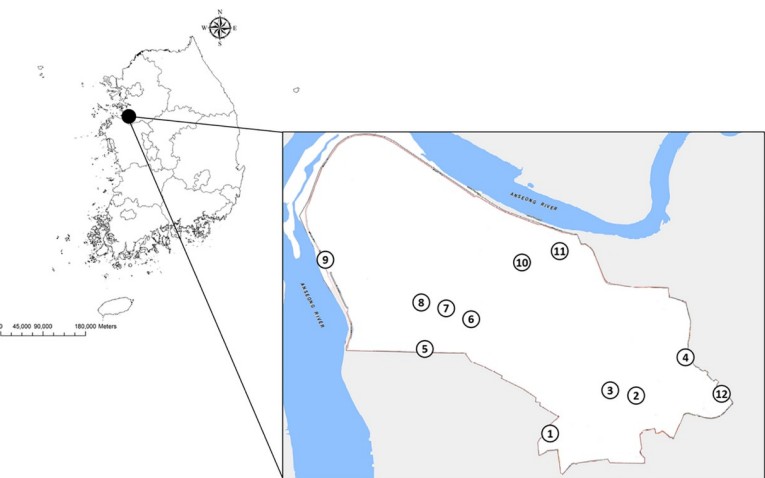

**Fig 1. General location of collection sites for mosquitoes collected by Mosquito Magnet® traps operated at 12 sites at USAG Humphreys, Pyeongtaek, Republic of Korea, during 2018 and 2019 mosquito season from May-October.**

## Materials and methods

### Characteristics of collection sites

USAG Humphreys is bordered by Anjeong-ri, a small village near Pyeongtaek-si (si = city), Gyeonggi province, wetland (rice) and other agriculture farming, and the Anseong River (Fig 1). USAG Humphreys, a US Army hub, currently houses about 30,000 military personnel and family members, while civilians and limited numbers of US military personnel reside in nearby cities/villages. In the interior of USAG Humphreys, there are small isolated areas of unmanaged grasses/herbaceous vegetation, small groves of trees and a central major drainage system, including water impoundments to reduce flooding. A total of 12 collection sites, adjacent to two child development care centers, family housing, the Brian Allgood D. Army Community Hospital, and water impoundments were selected to identify environmental and ecological factors corresponding to overall and site location for estimating mosquito-borne disease risks (Table 1, Fig 1).

**Table 1. Description of mosquito sampling sites on Camp Humphreys (see Fig 1 for general locations).**

| Sampling site | Latitude | Longitude | Description of location |
|---|---|---|---|
| Humphreys 1 | 36˚ 57' 01.74"N | 127˚ 01' 06.07"E | #1: Adjacent to family housing and rice paddies |
| Humphreys 2 | 36˚ 57' 22.36"N | 127˚ 01' 54.42"E | #2: Adjacent to the airfield |
| Humphreys 3 | 36˚ 57' 21.27"N | 127˚ 01' 42.51"E | #3: Border of a water retention basin |
| Humphreys 4 | 36˚ 57' 36.60"N | 127˚ 02' 25.86"E | #4: Adjacent to the Fire Station |
| Humphreys 5 | 36˚ 57' 38.39"N | 126˚ 59' 51.57"E | #5: Adjacent to a Child Development Center and rice paddies |
| Humphreys 6 | 36˚ 57' 56.74"N | 127˚ 00' 15.15"E | #6: Between Family Housing and permanent stream |
| Humphreys 7 | 36˚ 57' 59.91"N | 127˚ 00' 04.04"E | #7: Adjacent to Plaza and permanent stream |
| Humphreys 8 | 36˚ 57' 59.31"N | 126˚ 59' 54.17"E | #8: Adjacent to Family Housing, permanent stream, and water impoundment |
| Humphreys 9 | 36˚ 58' 21.37"N | 126˚ 58' 54.48"E | #9: Bordering the West Flood Gate and water retention basin |
| Humphreys 10 | 36˚ 58' 23.70"N | 127˚ 01' 08.98"E | #10: Bordering Water Detention Basin III |
| Humphreys 11 | 36˚ 58' 24.82"N | 127˚ 01' 09.09"E | #11: Bordering Water Detention Basin III and tank range |
| Humphreys 12 | 36˚ 57' 22.91"N | 127˚ 02' 45.63"E | #12: Forested park (Beacon Hill Park) and small pond |

## Mosquito traps and collections

Mosquito Magnet® traps (Liberty Plus model, Wood-stream Corp., Lititz, Pennsylvania, USA) using counter-flow technology and propane gas to produce heat and carbon dioxide ($CO_2$) as attractants were used to capture mosquitoes and other biting flies. This trap is widely used in entomological investigations since: (1) it does not have to be serviced daily, (2) it collects both day- and night-biting mosquitoes, (3) its performance results in high capture rates, (4) it captures a wide variety of mosquito species and other biting flies, and (5) it infrequently collects non-target flying insects, e.g., beetles and moths [14]. In addition, it was demonstrated that Mosquito Magnet traps were useful in mosquito control [15]. The Department of Public Works (DPW) collected the mosquitoes daily over a 24 hrs period for three consecutive days weekly from 1 May to 31 October in 2018 and from 15 May to 31 October in 2019. Attractants, e.g., octenol and citric acid were not used. The collection nets were removed from the traps from 07:00–08:00 each of the collection days, placed in a large Zip-lock bag and sealed to contain mosquitoes that escaped from the nets, and then placed in a Styrofoam cooler.

## Identification of mosquito species

DPW personnel transported the mosquitoes after each collection period to the Entomology Section, Forces Health Protection & Preventive Medicine, 65th Medical Brigade, and then placed in a -80˚C freezer. After 1 hour, the mosquitoes and other biting flies were carefully removed from each of the collection bags/Zip-lock bags, placed in Petri dishes with white filter paper, labeled with the trap site and date of collection, and then returned to the -80˚C freezer until identified. Mosquitoes in Petri dishes were removed from the -80˚C freezer and identified on a cold table to species or *Anopheles* Hyrcanus Group using morphological techniques [16, 17]. After identification, 1–50 mosquitoes were placed according to species (culicines) or *Anopheles* Hyrcanus Group and date and location of collection, in cryovials labeled with a unique collection number that corresponded to electronic data collection records. The database included the unique collection identification number and the date of collection, type of trap used, trap number (location), and the number of mosquitoes, by species, collected.

## Meteorological index

Meteorological indices were provided by the Korea Meteorological Administration. Daily mean temperatures and cumulative precipitation during 2018 and 2019 were provided by the Automated Surface Observing System, Pyeongtaek, ROK. The daily meteorological indices were matched with the mosquito collection database corresponding to the date of collection and other pertinent collection information.

## Experimental design

Mosquito collections at all sites were initiated on 15 and 3 May during 2018 and 2019, respectively, and concluded on when mosquito populations were very low on 31 October for both years. Mosquitoes were collected daily from each of the traps operated continuously for a period of three days/week. During 2018, adult mosquito control was coordinated with DPW for pesticide application using an ultra-low volume (ULV) fogger on 14 and 16 August, and 12, 19, and 27 September. Adult mosquito control was not conducted during 2019. Variables for insecticide application that directly affect the relative abundance of adult mosquitoes and indirectly affect larval populations were reflected in the model. Heat and $CO_2$, generated by the trap during its operation, were the only attractants.

## Modeling the empirical temperature association with mosquito abundance

The daily mean temperatures and cumulative precipitation from 7 to 13 days prior (lag 2 weeks) from the collection date were used for comparative analysis to estimate the temperature ranges of maximum abundance for each mosquito species. The Y-axis was the daily count of mosquitoes on day t ($x_t$), where $T_{mean}$ was the daily mean temperature, and $P_{cumulative}$ was the daily cumulative precipitation. Adult control applied with a ULV fogger, by season and year for the collection dates were adjusted for all models. Relative risk (RR) reduction was compared 1 day before ULV fogging application of pesticides and 1 day after ULV fogging were observed to assess the residual effect of adult mosquito control by mosquito species. The associations between the $Y_t$ and the $x_t$ axis using a generalized additive model (GAM) and a generalized linear model (GLM) with a quasi-Poisson family were analyzed.

$$Y_t \sim Quasi-Poisson(x_t)$$

$$Log(E(Y_t)) = \alpha + ns(T_{mean}) + ns(P_{cumulative}) + factor(year) + factor(season)$$

$$+factor(insecticide\ fogging)$$

where 'ns' is spline for flexible function on parameter $T_{mean}$ and $P_{cumulative}$.

A quasi-Poisson family was hypothesized that allowed for over-dispersion ("meaning that the variance of the outcome counts is higher than predicted under a Poisson distribution") [18]. The model selection criteria were based on "Akaike and Bayesian information criteria for models with over-dispersed outcomes fitted through quasi-likelihood" and quasi-Akaike Information Criterion (QAIC) values to determine the optimal choice [19]. In this model, the empirical mean temperatures (or median) and percentiles (e.g., 2.5th–97.5th) provide a range of temperature points and interval estimates for maximum mosquito abundance based on the bootstrap method [18]. In each model, the degree of freedom (df) was adjusted from 2 to 4, and after simulating the model, the lowest df for QAIC was selected and applied. After applying the selected df to the model and setting the knots in any of three cases (0.33–0.66, 0.10–0.50 and 0.50–0.90), the knot with the lowest QAIC was finally selected for Monte-Carlo simulation. Here, the knot is an arbitrary range to simulate training for fitting a smoothing spline. The empirical distribution of the maximum abundance of mosquito is asymmetric, in which case the choice of percentile can be adjusted according to the form of the empirical distribution. All statistical analyses were performed using the R version 3.5.3 (The R Foundation for Statistical Computing, Vienna, Austria). The level of statistical significance was set at $p = 0.05$ and 95% confidence intervals (CIs) were estimated for the point estimates. The ArcGIS 10.5 software (developed by Environmental System Research Institute located in Redlands, California, USA) was used to present a distribution for the relative abundance of mosquitoes. The shapefile data of the map was obtained from open source provided by the Ministry of Land, Infrastructure and Transport, Republic of Korea (data.nsdi.go.kr).

## Results

During the study period, mosquitoes were collected at 12 fixed sampling sites on Camp Humphreys (Table 1, Fig 1).

A total of 304,061 mosquitoes were collected using Mosquito Magnet® Liberty Plus model traps at 12 sites at USAG Humphreys over a period of 147 days, of which 162,231 mosquitoes were collected over a period of 73 days during 2018 and 141,830 mosquitoes were collected over a period of 74 days during 2019. The overall mean trap index (TI) (mean number of mosquitoes collected per trap per trap night) was 172.4 mosquitoes/trap/night, but varied by trap

site from 29.9 to 589.4 (Table 2). The trap indices for traps set near the border of Anjeong-ri and airfield were relatively low (range 29.9–37.1) compared to traps set adjacent to rice paddies (range 63.3–134.0), permanent streams (not always flowing, but with permanent water) (range 70.7–103.8), and those operated near water retention basins and the Anseong River (range 101.0–611.8).

*Culex tritaeniorhynchus* (Dyar) (43.1%) was the most frequently collected mosquito during 2018, followed by members of the *Anopheles* Hyrcanus Group (14.3%), *Mansonia uniformis* (Theobald) (13.2%), *Aedes vexans nipponii* (Theobald) (9.1), *Ae. lineatopennis* (Ludlow) (8.9%), *Cx. inatomii* (Kammimura and Wada) (5.5%), *Cx. bitaeniorhynchus* (Giles) (4.0%), and *Cx. pipiens* (L.) (1.5%), while the remaining species only accounted for 0.3% (range <0.1– 0.1%) (Table 3). However, in 2019, *Mn. uniformis* (57.5%) was the most frequently collected mosquito, followed by members of the *Anopheles* Hyrcanus Group (12.9%), *Cx. inatomii* (12.1%), *Ae. vexans nipponii* (8.1%), *Cx. pipiens* (2.5%), *Cx. bitaeniorhynchus* (2.2%), *Ae. lineatopennis* (8.9%), and *Cx. tritaeniorhynchus* (1.1%), while the remaining mosquito species only accounted for 0.5% (range <0.1–0.2%). Overall, *Mn. uniformis* (33.7%) was the most frequently collected mosquito, followed by *Cx. tritaeniorhynchus* (23.5%), *Anopheles* Hyrcanus Group (13.7%), *Ae. vexans nipponii* (8.6%), and *Ae. lineatopennis* (5.7%). The greatest differences between 2018 and 2019 were observed for *Coquilletidia ochracea* (Theobald) (+98.1%), followed by *Cx. tritaeniornynchus* (-95.6%), *Ochlerotatus dorsalis* (Meigen) (+87.8%), *Ae. lineatopennis* (-67.2%), *Mn. uniformis* (+58.5%) (Table 3).

The distribution of the mosquitoes at USAG Humphreys, by year, is shown in Fig 2. The ratio of mosquito species collected by the sampling sites was different by year. Among total mosquito populations, *Cx. tritaeniorhynchus* was collected at a ratio of 30–60% in 2018, but less than 10% at all sites in 2019. *Mn. uniformis* was collected at a high proportion in 2019 compared to 2018, and over 70% at some sites. While the seasonality of mosquito species was variable for 2018 and 2019, some species were collected predominantly during the early-mid mosquito season, e.g., *Ae. vexans nipponii*, *Anopheles* Hyrcanus Group, *Cx. pipiens*, *Cx. inatomii*, while others, e.g., *Ae. lineatopennis*, *Cx. tritaeniorhynchus*, and *Mn. uniformis*, were

**Table 2. Total number of mosquitoes collected at USAG Humphreys by Independence⒭ model Mosquito Magnet⒭ traps during 2018 and 2019.**

| Sampling site[a] | 15 May-31 Oct 2018 | | | 3 May-31 Oct 2019 | | | Total | Overall Mean TI[b] |
|---|---|---|---|---|---|---|---|---|
| | Number trap nights | Total collected | Mean TI[b] | Number trap nights | Total collected | Mean TI[b] | | |
| Humphreys 1 | 73 | 5,061 | 69.3 | 74 | 4,239 | 57.3 | 9,300 | 63.3 |
| Humphreys 2 | 73 | 3,747 | 51.3 | 74 | 1,711 | 23.1 | 5,458 | 37.1 |
| Humphreys 3 | 73 | 17,500 | 239.7 | 74 | 4,978 | 67.3 | 22,478 | 152.9 |
| Humphreys 4 | 73 | 1,953 | 26.8 | 74 | 2,754 | 37.2 | 4,707 | 32.0 |
| Humphreys 5 | 73 | 8,687 | 119 | 74 | 11,015 | 148.9 | 19,702 | 134.0 |
| Humphreys 6 | 73 | 5,539 | 75.9 | 74 | 4,861 | 65.7 | 10,400 | 70.7 |
| Humphreys 7 | 73 | 10,253 | 140.5 | 74 | 5,006 | 67.6 | 15,259 | 103.8 |
| Humphreys 8 | 73 | 3,280 | 44.9 | 74 | 11,563 | 156.3 | 14,843 | 101.0 |
| Humphreys 9 | 73 | 62,196 | 852 | 74 | 24,450 | 330.4 | 86,646 | 589.4 |
| Humphreys 10 | 73 | 8,541 | 117 | 74 | 12,411 | 167.7 | 20,952 | 142.5 |
| Humphreys 11 | 73 | 34,289 | 469.7 | 74 | 55,639 | 751.9 | 89,928 | 611.8 |
| Humphreys 12 | 73 | 1,185 | 16.2 | 74 | 3,203 | 43.3 | 4,388 | 29.9 |
| Total | 876 | 162,231 | 185.2 | 888 | 141,830 | 159.7 | 304,061 | 172.4 |

[a] One Mosquito Magnet, Independence model, was operated 3 nights/week May-October at each of the collection sites.

[b] TI (trap index) = mean number of mosquitoes collected per trap night.

**Table 3. Total number and percentage of mosquitoes collected by species at USAG Humphreys using Mosquito Magnet® Independence® model traps during 2018 and 2019 mosquito season from May-October.**

| Species | 2018 | | 2019 | | Total | % | (+) / (-)[b] |
|---|---|---|---|---|---|---|---|
| | N | % | N | % | | | (%) |
| *Aedes albopictus* | 82 | 0.05 | 232 | 0.16 | 314 | 0.10 | +47.8 |
| *Aedes alboscutellatus* | 102 | 0.06 | 36 | 0.03 | 138 | 0.05 | -47.8 |
| *Aedes lineatopennis* | 14,510 | 8.94 | 2,846 | 2.01 | 17,356 | 5.71 | -67.2 |
| *Aedes vexans nipponii* | 14,720 | 9.07 | 11,483 | 8.10 | 26,203 | 8.62 | -12.4 |
| *Anopheles* Hyrcanus Group[a] | 23,271 | 14.34 | 18,277 | 12.88 | 41,548 | 13.64 | -12.0 |
| *Anopheles sineroides* | 1 | <0.01 | 0 | 0.0 | 1 | <0.01 | -100.0 |
| *Armigeres subalbatus* | 17 | 0.01 | 15 | 0.01 | 32 | 0.01 | -6.3 |
| *Culex pipiens* | 2,486 | 1.53 | 3,519 | 2.48 | 6,005 | 1.98 | +17.2 |
| *Culex bitaeniorhynchus* | 6,480 | 3.99 | 3,163 | 2.23 | 9,643 | 3.17 | -34.4 |
| *Culex inatomii* | 8,952 | 5.52 | 17,105 | 12.06 | 26,057 | 8.57 | +31.3 |
| *Culex tritaeniorhynchus* | 69,914 | 43.10 | 1,575 | 1.11 | 71,489 | 23.51 | -95.6 |
| *Culex orientalis* | 28 | 0.02 | 128 | 0.09 | 156 | 0.05 | +64.1 |
| *Culiseta nipponica* | 105 | 0.06 | 84 | 0.06 | 189 | 0.06 | -11.1 |
| *Coquillettidia ochracea* | 15 | <0.01 | 1,593 | 1.12 | 1,608 | 0.53 | +98.1 |
| *Mansonia uniformis* | 21,345 | 13.16 | 81,521 | 57.48 | 102,866 | 33.83 | +58.5 |
| *Ochlerotatus dorsalis* | 11 | <0.01 | 170 | 0.12 | 181 | 0.06 | +87.8 |
| *Ochlerotatus koreicus* | 192 | 0.12 | 83 | 0.06 | 275 | 0.09 | -39.6 |
| Total | 162,231 | 100.0 | 141,830 | 100.0 | 304,061 | 100.0 | -6.7 |

[a] *Anopheles* Hyrcanus Group includes *An. sinensis* s.s., *An. belenrae*, *An. lesteri*, *An. kleini*, and *An. pullus* that cannot be identified by morphological techniques.

[b] Percent increase (+)/decrease (-) in the total number of mosquitoes collected during 2018 compared to the total number of mosquitoes collected during 2019. (+) = proportion of increase during 2019 compared to 2018 and (-) = proportion of decrease during 2019 compared to 2018.

predominately collected during the mid- to late-season (Table 4). Overall, in 2018, the number of mosquitoes collected in July and August demonstrated small increases in numbers when the daily mean temperatures were highest (Fig 3A).

The time-series distribution of mosquitoes daily from all 12-collection sites, mean weekly numbers of mosquitoes collected, daily mean temperatures, the daily mean precipitation is shown in Fig 3. There were no significant differences in the overall and corresponding daily mean temperatures in 2018 (19.7˚C) and 2019 (19.8˚C) during the mosquito surveillance season. Similarly, the maximum daily mean temperatures for 2018 (32.4˚C) and 2019 (31.7˚C) was only slightly higher during 2018. While the daily mean precipitation was 4.1 times higher in 2019 (19.9 mm) than in 2018 (4.8 mm), the maximum daily mean precipitation for 2018 (124.8 mm) and 2019 (108.0 mm) were similar.

Adult mosquito control using ULV fogging application is widely used by the US military for reducing adult mosquito populations. The impact of application of insecticide using a ULV fogger was minimally effective, with a decrease in the total numbers of mosquitoes collected on day 1 post-application (Log[RR]: -0.27 [95% CI: -0.43--0.11], $p<0.05$). We also assessed the short-term effectiveness of this factor using GLM (Fig 4) by species. ULV fogging contributed to the greatest reduction of *Ae. lineatopennis* (Log[RR]: -2.08 [95% CI: -2.78--1.7], $p<0.05$) and *Ae. albopictus* (Log[RR]: -1.75 [95% CI: -2.16--1.34], $p<0.05$) numbers of mosquitoes collected after 1 day.

The exposure-response relationship between the numbers of mosquitoes and daily mean temperatures, adjusted for the cumulative precipitation of the 2-week lag effect using the total number of mosquitoes and the number of species collected in 2018 and 2019 is shown in Fig 5.

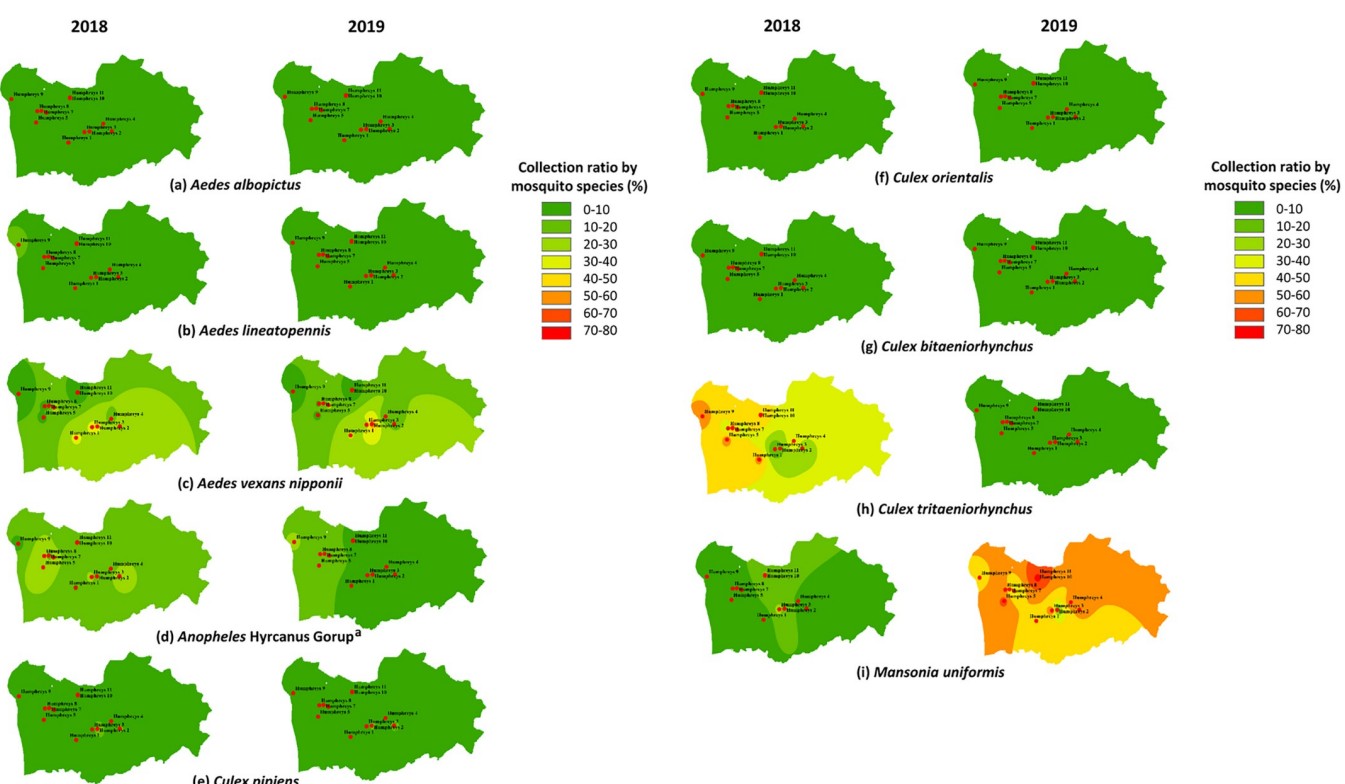

**Fig 2. Geographical distribution of mean numbers of mosquitoes collected for selected species at USAG Humphreys, Pyeongtaek, Republic of Korea using Mosquito Magnet® traps during 2018 and 2019.** *Ratio was calculated as the number of mosquitoes collected by species/the number of total mosquitoes. [a]*Anopheles* Hyrcanus Group includes *An. sinensis* s.s., *An. belenrae*, *An. lesteri*, *An. kleini*, and *An. pullus* that cannot be identified by morphological techniques.

Overall, the mosquito population declined or did not increase any more than a certain temperature (threshold temperature) for most species. In particular, *Cx. orientalis* (Edwards) (Fig 5G), *Cx. bitaeniorhynchus* (Giles) (Fig 5H) and *Mn. uniformis* (Fig 5J) showed a strong significant relationship for specific temperature ranges. However, in the case of *Ae. vexans nipponii* (Fig 5D), the population continued to increase as temperatures increased without a specific temperature point threshold.

The empirical mean temperatures in combination with maximum abundance for selected species were estimated in the range of 2.5th–97.5th using Monte-Carlo simulation (Table 5).

Overall, the empirical mean temperatures during maximum abundance was 22.7°C (2.5th–97.5th: 21.7°C–23.8°C) and were statistically significant. However, the maximum abundance compared to the empirical mean temperatures varied by species, e.g., the empirical mean temperatures for *Ae. albopictus* was 24.6°C (2.5th–97.5th: 22.3°C–25.6°C), *Cx. tritaeniorhynchus* was 24.3°C (2.5th–97.5th: 21.9°C–26.3°C), *Ae. vexans nipponii* was 24.3°C (2.5th–97.5th: 21.9°C–25.2°C), *Ae. lineatopennis* was 23.2°C (2.5th–97.5th: 20.9°C–28.5°C), *Cx. pipiens* was 22.6°C (2.5th–97.5th: 21.9°C–25.2°C) and *Anopheles* Hyrcanus Group was 22.4°C (2.5th–97.5th: 21.5°C–23.8°C). The maximum abundance temperature range was higher for *Cx. orientalis* at 26.6°C (2.5th–97.5th: 22.2°C–27.5°C) and were similar for both *Cx. bitaeniorhynchus* and *Mn. uniformis* at 27.0°C (2.5th–97.5th: 23.7°C–28.3°C) and 27.0°C (2.5th–97.5th: 26.3°C–27.6°C), respectively. While most mosquito species showed the maximum abundance distributions at different temperature ranges, the maximum abundance distribution for *Ae. lineatopennis* was

**Table 4. Total number of mosquitoes collected, by month, at USAG Humphreys using Independence® model Mosquito Magnet® traps during 2018 and 2019.**

| Species | 2018 | | | | | | | 2019 | | | | | | |
|---|---|---|---|---|---|---|---|---|---|---|---|---|---|---|
| | May | Jun | Jul | Aug | Sep | Oct | Total | May | Jun | Jul | Aug | Sep | Oct | Total |
| *Aedes albopictus* | 3 | 17 | 23 | 13 | 24 | 2 | 82 | 0 | 11 | 9 | 39 | 153 | 20 | 232 |
| *Aedes alboscutellatus* | 0 | 0 | 2 | 3 | 96 | 1 | 102 | 0 | 0 | 0 | 0 | 34 | 2 | 36 |
| *Aedes lineatopennis* | 52 | 102 | 2,255 | 383 | 11,338 | 380 | 14,510 | 0 | 181 | 62 | 1,533 | 993 | 77 | 2,846 |
| *Aedes vexans nipponii* | 3,849 | 4,183 | 3,633 | 2,243 | 756 | 56 | 14,720 | 3,539 | 2,760 | 2,404 | 1,887 | 831 | 62 | 11,483 |
| *Anopheles* Hyrcanus Group[a] | 1,284 | 7,927 | 4,535 | 5,957 | 3,175 | 393 | 23,271 | 2,354 | 7,461 | 5,174 | 1,912 | 1,287 | 89 | 18,277 |
| *Anopheles sineroides* | 0 | 0 | 0 | 1 | 0 | 0 | 1 | 0 | 0 | 0 | 0 | 0 | 0 | 0 |
| *Armigeres subalbatus* | 0 | 0 | 7 | 6 | 3 | 1 | 17 | 0 | 3 | 1 | 6 | 4 | 1 | 15 |
| *Culex pipiens* | 225 | 1,037 | 412 | 673 | 113 | 26 | 2,486 | 231 | 871 | 1,901 | 256 | 194 | 66 | 3,519 |
| *Culex bitaeniorhynchus* | 20 | 528 | 1,809 | 3,577 | 546 | 0 | 6,480 | 1 | 42 | 2,269 | 557 | 290 | 4 | 3,163 |
| *Culex inatomii* | 1,192 | 4,434 | 851 | 1,760 | 670 | 45 | 8,952 | 2,738 | 6,357 | 7,280 | 651 | 69 | 10 | 17,105 |
| *Culex tritaeniorhynchus* | 1 | 69 | 1,665 | 38,051 | 28,841 | 1,287 | 69,914 | 0 | 0 | 7 | 297 | 1,142 | 129 | 1,575 |
| *Culex orientalis* | 1 | 0 | 6 | 19 | 2 | 0 | 28 | 0 | 1 | 26 | 66 | 35 | 0 | 128 |
| *Culiseta nipponica* | 0 | 74 | 21 | 9 | 1 | 0 | 105 | 0 | 3 | 68 | 11 | 2 | 0 | 84 |
| *Coquillettidia ochracea* | 0 | 1 | 0 | 1 | 13 | 0 | 15 | 0 | 44 | 467 | 468 | 595 | 19 | 1,593 |
| *Mansonia uniformis* | 4 | 596 | 1,818 | 10,576 | 8,263 | 88 | 21,345 | 362 | 7,417 | 18,834 | 37,005 | 17,635 | 268 | 81,521 |
| *Ochlerotatus dorsalis* | 4 | 0 | 3 | 1 | 3 | 0 | 11 | 3 | 22 | 6 | 29 | 107 | 3 | 170 |
| *Ochlerotatus koreicus* | 7 | 33 | 139 | 9 | 3 | 1 | 192 | 16 | 12 | 27 | 14 | 9 | 5 | 83 |
| Total | 6,642 | 19,001 | 17,179 | 63,282 | 53,847 | 2,280 | 162,231 | 9,244 | 25,185 | 38,535 | 44,731 | 23,380 | 755 | 141,830 |

[a]*Anopheles* Hyrcanus Group includes *An. sinensis* s.s., *An. belenrae*, *An. lesteri*, *An. kleini*, and *An. pullus* that cannot be identified by morphological techniques.

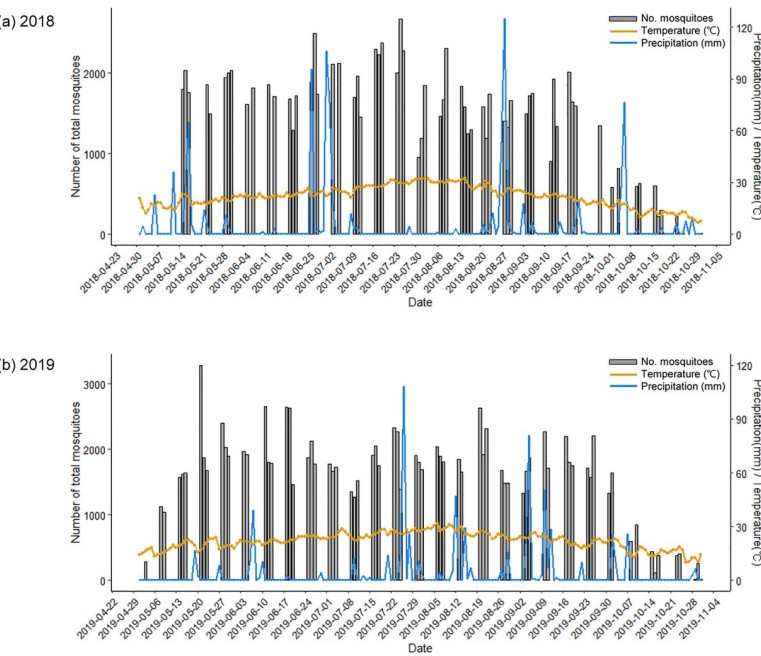

**Fig 3.** Total numbers of mosquitoes collected daily using Mosquito Magnet® traps operated at 12 collection sites at USAG Humphreys, and daily mean temperature and daily cumulative precipitation during 2018 (a) and 2019 (b). The yellow line is daily mean temperatures (˚C) and the blue line is the daily mean precipitation (mm) from 22 April through 4 November.

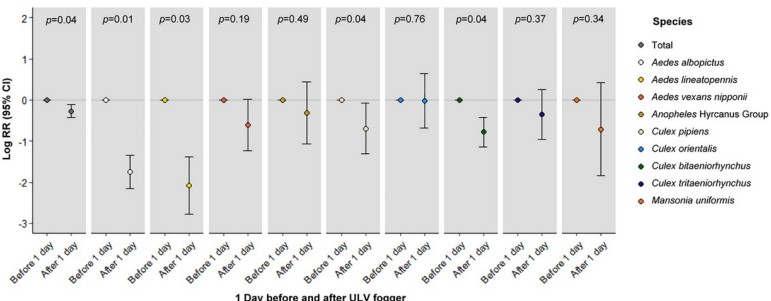

**Fig 4. Comparison the mean numbers (range) of mosquitoes collected daily using Mosquito Magnet® traps operated at 12 collection sites at USAG Humphreys 1 day before and the following day after pesticide application using an ULV fogger during 2018.** *Anopheles* Hyrcanus Group includes *An. sinensis* s.s., *An. belenrae, An. lesteri, An. kleini,* and *An. pullus* that cannot be identified by morphological techniques. *Adjusted for a lag of 2-weeks compared to daily mean temperatures and cumulative precipitation, season. RR, relative risk; CI, confidence interval.

not significantly different. Insufficient numbers of some species were collected to determine maximum abundance distributions.

## Discussion

USAG Humphreys is bordered by a small city/village on one side, wetland rice agriculture and farming on two sides, and a major river bordered by tall grasses and low lying areas that flood on the other side. Internally, USAG Humphreys is composed of various environmental factors ranging from urban-like areas to a central stream associated with multiple water impoundments bordered by tall grasses and internally with emergent and floating vegetation that is conducive for larval development. Mosquito larvae were collected throughout the central area and flood zones that contributed to large numbers of adult mosquitoes, including primary and secondary vectors of JEV and members of the *Anopheles* Hyrcanus Group that transmit malaria. Adult and larval surveillance provides information for potential disease risks and for implementing vector control measures. The number of traps that are highly efficient for collecting all species of mosquitoes increased the potential to observe mosquito species-specific responses modified by the habitat environment factors.

The distribution of adult mosquito populations is dependent upon seasonal climatic and ecological factors that affect larval growth and development. In the ROK, mosquito populations gradually become more active from May-late August/early September, and decreased thereafter through October when the mean temperatures decreased significantly (Fig 3). This supports the rationale that a certain temperature or higher is essential for eclosion and development of mosquito larvae to the adult stage. In this study, no seasonality was observed, because mosquito monitoring was performed only during the most active period. However, the relationship between relative mosquito abundance and daily mean temperatures for vectors that are of medical importance and nuisance bitters were determined by conducting daily collections throughout the mosquito season using highly efficient Mosquito Magnet® traps at USAG Humphreys during 2018 and 2019. These traps not only provided a good estimate of mosquito abundance, but also demonstrated effective control of biting mosquitoes [15]. Thus, the use of Mosquito Magnet traps® provided for our ability to identify increasing disease risks over time by evaluating the range of the maximum mosquito abundance [20]. These data provided evidence that support modeling studies based on previous epidemiological data [21]. Findings also demonstrated consistent results with previous studies that showed that the occurrence of mosquito populations increases over time as temperature increases, but also

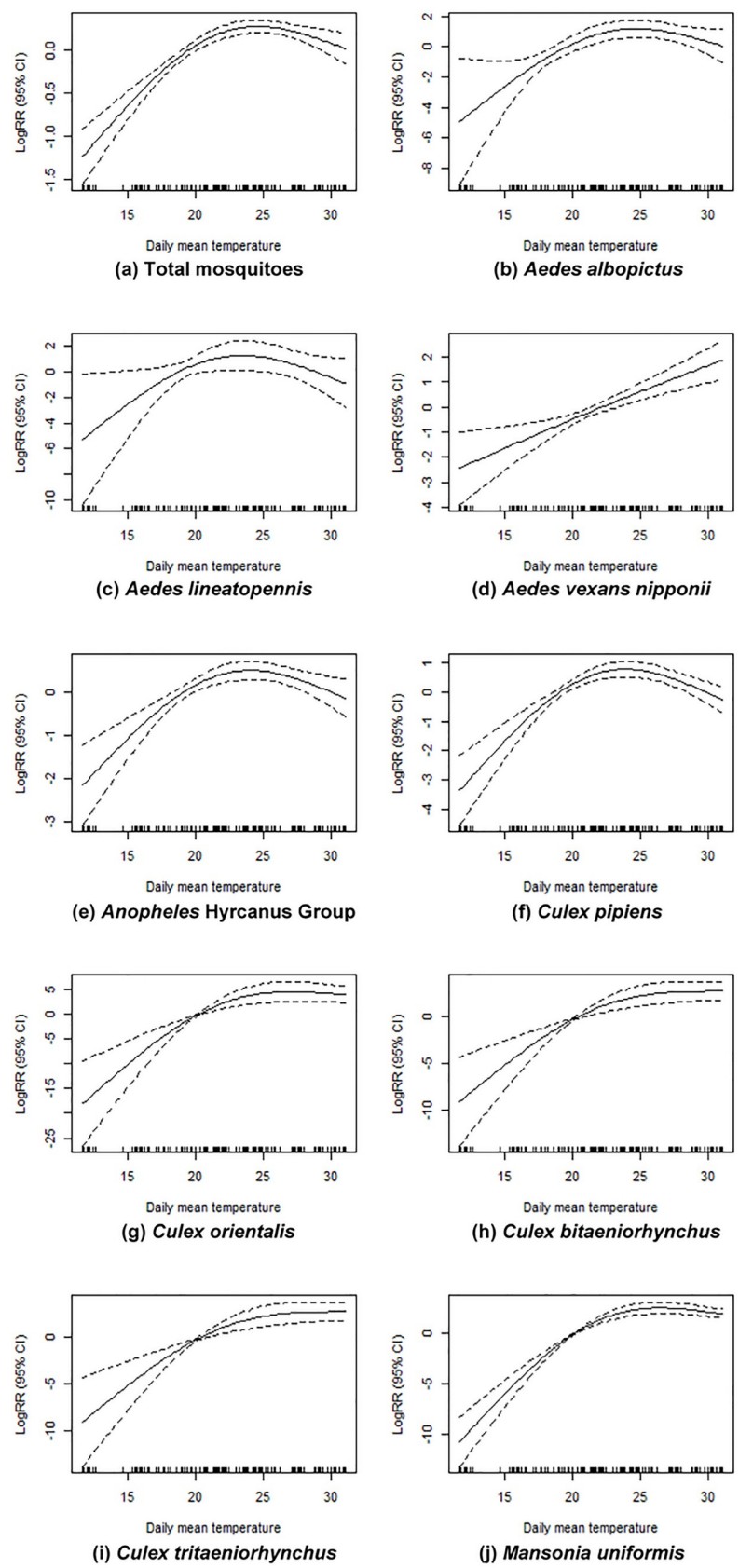

**Fig 5. Relationship of Generalized Additive Model (GAM) between the abundance of selected mosquito species and daily mean temperatures, adjusted for the cumulative precipitation based on a 2-week lag effect for mosquitoes collected during the 2018 and 2019 mosquito season using Mosquito Magnet® traps operated at 12 collection sites at USAG Humphreys.** The solid line is the estimated log (relative risk) as density of mosquitoes by species and dashed line is the 95% confidence interval. *Adjusted for a lag 2-week lag for cumulative precipitation, year, season, and presence of insecticide fogging. RR, relative risk.

decreases above certain temperatures [13, 22–24]. Therefore, it is important to identify optimum temperature ranges at which the maximum mosquito abundance occurs for each species to effectively implement vector control measures. An estimate of the nonlinear relationship between mosquito abundance and daily average temperatures, based on the lag effect that takes into account the period of 2 weeks from egg to adult before the collection date, was performed [25, 26]. In subsequent years, these data can be used to determine the effectiveness of vector control measures, e.g., different types of adult fogging measures (e.g., ULV or thermal fogging) or larval control (e.g., use of Bti) to reduce disease risks.

*Culex orientalis* and *Cx. pipiens*, which were collected in relatively high numbers, were in the same *Culex* group, but the maximum abundance temperatures differed by 4°C, 26.6°C and 22.6°C, respectively. A previous study conducted in the same region showed that the JEV Genotype V was detected in *Cx. bitaeniorhynchus* collected near the demilitarized zone (DMZ) [16, 27] and later at other sites, including USAG Humphreys in 2018 [28, J Hang Personal Communication]. A viral genome associated with the Chaoyang virus, a potential anthropod-specific flavivirus, was isolated from *Ae. vexans nipponii* also accounted for a significant proportion of mosquitoes collected [29]. However, *Ae. vexans nipponii* is a flood water mosquito and relative abundance increased linearly with no observed threshold temperature, and our model was unable to provide the maximum density temperature range. Thus, populations of *Ae. vexans nipponii* may be more related to flooding of larval habitats than temperatures during the mosquito season. A *Rickettsia* sp. was recently isolated from *Mn. uniformis* collected near the DMZ, but has not been detected in the large numbers of *Mn. uniformis* collected and assayed using next generation sequencing (NGS) at USAG Humphreys, and which accounted for the most of the mosquitoes collected during 2019 [30]. Additionally, Getah virus, which is of veterinary importance, was identified in mosquitoes collected near the DMZ,

**Table 5. Estimated empirical mean temperature for maximum abundance of mosquitoes by species collected using Independence® model Mosquito Magnet® traps.**

| Species | df[a] | QAIC[b] | Temperature (°C)[c] | 2.5th–97.5th | *p*-value |
|---|---|---|---|---|---|
| Total mosquitoes | 3 | 37642.2 | 22.7 | 21.7–23.8 | <0.0001 |
| *Aedes albopictus* | 2 | 799.3 | 24.6 | 22.3–25.6 | <0.05 |
| *Aedes lineatopennis* | 2 | 57459.4 | 23.2 | 20.9–28.5 | 0.0934 |
| *Anopheles* Hyrcanus Group[d] | 4 | 33028.1 | 22.4 | 21.5–23.8 | <0.0001 |
| *Culex pipiens* | 3 | 6287.5 | 22.6 | 21.9–25.2 | <0.0001 |
| *Culex orientalis* | 3 | 399.6 | 26.6 | 22.2–27.5 | <0.0001 |
| *Culex bitaeniorhynchus* | 4 | 11707.5 | 27.0 | 23.7–28.3 | <0.0001 |
| *Culex tritaeniorhynchus* | 2 | 22814.3 | 24.3 | 21.9–26.3 | <0.05 |
| *Mansonia uniformis* | 2 | 95176.2 | 27.0 | 26.3–27.6 | <0.0001 |

[a] df, degree of freedom; QAIC, quasi-Akaike Information Criterion. The df was selected for the minimum QAIC.

[b] Adjusted for the lag 2-week cumulative precipitation, year, season, and presence of insecticide spraying.

[c] Estimated empirical mean temperatures and percentiles (2.5th–97.5th) based on the bootstrap method in Monte-Carlo simulation with the selected the lowest QAIC of knot.

[d] *Anopheles* Hyrcanus Group includes *An. sinensis* s.s., *An. belenrae*, *An. lesteri*, *An. kleini*, and *An. pullus* that cannot be identified by morphological techniques.

but not in mosquitoes collected at USAG Humphreys. While *Rickettsia* spp. and Getah virus has not been detected at USAG Humphreys, the Walter Reed Army Institute of Research has identified novel viruses using NGS in mosquitoes collected at USAG Humphreys [31, J. Hang personal communication]. The significance of these findings are not well understood, with most viruses assumed to be mosquito-specific and not of veterinary or medical importance. Therefore, to monitor vector control measures and limit the potential impact of viruses and other pathogens, it is very important to identify the timing of peak adult abundance to determine the effectiveness for reducing disease risks to military and civilian communities [20].

There are some limitations to our study. First, mosquitoes were collected only three times a week due to limited manpower to collect, identify, and process mosquitoes for pathogen detection. Mosquitoes were collected from Tuesday-Thursday during optimum and suboptimal weather conditions (e.g., rain) and may pose a potential bias to mosquito abundance for days not collected throughout the week. Second, only 2 years of collected data were analyzed. Using data for additional years of varying annual and seasonal climatic variations would provide better estimates of the effect of empirical temperatures and precipitation. Third, female mosquitoes, which are blood feeders, provide a good estimate for developing disease risk assessments and potential for the transmission of pathogens of medical importance. Therefore, it is imperative to understand the distribution of female mosquitoes and determine their relative abundance based on empirical temperatures to estimate temperature ranges that have a direct effect on the potential for increased transmission of mosquito-borne infectious diseases.

Nevertheless, our investigation has several strengths. First, daily mosquito collections were conducted consecutively 3-days/week throughout the mosquito season at 12 fixed monitoring sites. This was essential for controlling spatial-related factors and identifying time-series changes in mosquito abundance throughout the mosquito season. In addition, studies that conduct daily mosquito collections through the management of high-level collection equipment in the ROK are unusual. The national mosquito monitoring data currently held by the Korea Centers for Disease Control and Prevention (K-CDC) holds several years of database for mosquito collections conducted since the 1990s. However, the weekly mosquito collections were conducted for two consecutive days at disperse non-fixed monitoring sites, rather than annual fixed locations. To control the spatial variables that provide a relationship between climatic factors and relative mosquito abundance that accounts for the mosquito ecological cycle, continuous fixed monitoring sites are essential. Secondly, mosquitoes were collected at temperature ranges throughout the mosquito season, including increasing and decreasing temperatures that affect adult mosquito activity. These data provide a suitable estimate of temperature ranges, including the highest abundance of adult mosquito populations throughout the year and controls confounders that only estimate data within a selected temperature range. If data only includes information for a selected period of the mosquito season, there is a risk that the maximum mosquito abundance temperatures may actually appear higher or lower than our estimate. Third, we assessed the short-term effects of five events during 2018 following the application of pesticides by ULV fogging for adult mosquito control. Data showed that there were significant differences in catch rates following the application, although there was a temporary decrease on day 1 post-application. The residual effect of pesticide application is an important factor reducing vector populations by observing the time-series changes of the adult mosquito population [32]. Thus, other methods, e.g., using thermal foggers, changing pesticide application timing (e.g., at 10:00 PM), application coverage, and changing pesticides due to potential resistance, are essential to evaluate to reduce the potential for transmission of pathogens affecting human health.

Lastly, there is a strength that the temperature ranges of maximum abundance for selected mosquito species was suggested based on the mosquito ecological development cycle. Previous

studies examining the simple association between climatic factors and mosquito abundance have identified the relationships of numbers of collected mosquitoes with correlation coefficients, but we have the advantage that the temperature estimates were estimated with point estimates and temperature ranges of 2.5th–97.5th percentiles. Although this figure may differ slightly depending on climatic and environmental factors suitable for mosquito habitat factors for other areas in the ROK, the data provides information for temperatures suitable for mosquito development at USAG Humphreys.

Temperature and precipitation are the most important factors in mosquito population abundance. Higher temperatures decrease the development time of mosquito larvae, resulting in increased numbers of generations and higher adult populations. In general, the prevalence of mosquito-borne diseases follows a rapid increase in vector populations and follow an increased potential of disease among human populations with subsequent increases in the incidence of pathogens in mosquitoes [33]. As mean temperatures potentially increase annually due to the effects of global warming, there is the potential for increased disease transmission of local pathogens and the potential for the introduction of pathogens from other areas of the world that may become established. Thus, these data are necessary to determine the pattern of relative mosquito abundance that affects epidemiologic data and to determine the prevalence of mosquito-borne infectious diseases.

## Conclusion

The empirical mean temperatures distributed at the maximum abundance for selected mosquito species using mosquito data collected daily during 2018 and 2019 mosquito season were determined for USAG Humphreys, Pyeongtaek, Republic of Korea. This data provided information important for the implementation of control measures to reduce vector populations and the potential for transmission of pathogens of medical importance to US military and civilian populations, including surrounding areas.

## Acknowledgments

We thank COL Derek C. Cooper, Commander, 65th Medical Brigade and COL Jon Allison, Chief, Force Health Protection and Preventive Medicine, 65th Medical Brigade for their support of vector-borne disease surveillance at US military installations/training sites and associated villages in the Republic of Korea.

## Author Contributions

**Conceptualization:** Myung-Jae Hwang, Heung-Chul Kim, Hae-Kwan Cheong.

**Data curation:** Heung-Chul Kim, Sung-Tae Chong.

**Formal analysis:** Myung-Jae Hwang, Kisung Sim.

**Funding acquisition:** Heung-Chul Kim.

**Investigation:** Terry A. Klein, Sung-Tae Chong.

**Methodology:** Heung-Chul Kim, Kisung Sim, Yeonseung Chung, Hae-Kwan Cheong.

**Project administration:** Heung-Chul Kim.

**Resources:** Heung-Chul Kim.

**Software:** Myung-Jae Hwang, Kisung Sim.

**Supervision:** Heung-Chul Kim, Terry A. Klein, Yeonseung Chung, Hae-Kwan Cheong.

**Validation:** Heung-Chul Kim, Hae-Kwan Cheong.

**Writing – original draft:** Myung-Jae Hwang.

**Writing – review & editing:** Heung-Chul Kim, Terry A. Klein, Yeonseung Chung, Hae-Kwan Cheong.

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
