## [Decision Letter · Decision Letter 0]

23 Jul 2020

PONE-D-20-17361

Comparison of climatic factors on mosquito abundance at US Army Garrison Humphreys, Republic of Korea

PLOS ONE

Dear Dr. Cheong,

Thank you for submitting your manuscript to PLOS ONE. After careful consideration, we feel that it has merit but does not fully meet PLOS ONE’s publication criteria as it currently stands. Therefore, we invite you to submit a revised version of the manuscript that addresses the points raised during the review process.

Specifically, both reviewers raise issues with the time lag methodology. Please address these concerns carefully and supply additional analysis where appropriate.  

We look forward to receiving your revised manuscript.

Kind regards,

Silvie Huijben

Academic Editor

PLOS ONE

Journal Requirements:

3. We note that Figure 1 in your submission contains map/satellite images which may be copyrighted.

We require you to either (a) present written permission from the copyright holder to publish this figure specifically under the CC BY 4.0 license, or (b) remove the figure from your submission:

b. If you are unable to obtain permission from the original copyright holder to publish this figure under the CC BY 4.0 license or if the copyright holder’s requirements are incompatible with the CC BY 4.0 license, please either i) remove the figure or ii) supply a replacement figure that complies with the CC BY 4.0 license. Please check copyright information on all replacement figures and update the figure caption with source information. If applicable, please specify in the figure caption text when a figure is similar but not identical to the original image and is therefore for illustrative purposes only.

4. We noted in your submission details that a portion of your manuscript may have been presented or published elsewhere:

'This paper was uploaded as a preprint in the Research Square (https://www.researchsquare.com/article/rs-28172/v1) on 12 May 2020. However, it is not submitted or under consideration of publication in other journals or media. '

Please clarify whether this publication was peer-reviewed and formally published.

If this work was previously peer-reviewed and published, in the cover letter please provide the reason that this work does not constitute dual publication and should be included in the current manuscript.

5. Please amend the manuscript submission data (via Edit Submission) to include author Yeonseung Chung.

Reviewers' comments:

Reviewer's Responses to Questions

**Comments to the Author**

1. Is the manuscript technically sound, and do the data support the conclusions?

Reviewer #1: Partly

Reviewer #2: Yes

2. Has the statistical analysis been performed appropriately and rigorously? 

Reviewer #1: Yes

Reviewer #2: Yes

3. Have the authors made all data underlying the findings in their manuscript fully available?

Reviewer #1: No

Reviewer #2: Yes

4. Is the manuscript presented in an intelligible fashion and written in standard English?

Reviewer #1: Yes

Reviewer #2: Yes

5. Review Comments to the Author

Reviewer #1: The authors collected a valuable dataset containing mosquito diversity and abundance information from an army base in the Republic of Korea (May-Oct 2018, May-Oct 2019). Mean daily temperature and daily precipitation data were obtained from a nearby weather station, and empirical mean temperature ranges for maximum abundance were determined for each mosquito species.

Again, the dataset is large and unique, but I do have some concerns regarding the analysis:

1) A time lag of 2 weeks was included in the model. However, do larval development times vary per species? And did you consider the potential age range of the adult population? A 2-week time lag assumes you only collect very young adults in the traps. How would the temperature-abundance relation look if different time lags were included (which ideally take into account the larval development times and mean/median age of the adult population for each species, if you can find literature on that)?

2) On the topic of ‘time lags’: adult control applied with a ULV fogger was adjusted for all models, looking at a period of 1-4 days post-intervention. I am not familiar with the study area, but given the description it may be impossible to see an effect, given the amount of breeding sites (and thus mosquitoes) surrounding the army base. In addition, an effect is expected (i) after the next gonotrophic cycle has been completed, and not the next day, for many species, and (ii) 1-2 weeks later given the reduced numbers of eggs/larvae that go into the system. This makes me wonder if the current analysis is valid (to conclude ULV is not effective).

3) Line 342 “Third, there is a limitation that classification by sex of mosquitoes was not possible according to the mosquito species, since the Mosquito Magnet® infrequently collects male mosquitoes”. I do not fully understand this sentence. Why was it not possible to distinguish the males from the females? Males have more fine hairs on their antennae, correct? This sentence came a bit ‘out of the blue’ for me, as I assumed you were showing results for the females, as they are relevant for disease transmission. How does this affect your results?

4) In the discussion I am missing a section on climatic data (such as extremes, seasonal and daily fluctuations, micro-climatic conditions) and mosquito behaviors (thermal avoidance, resting biology).

Minor comments:

- Why the May-Oct periods? Is this a particular season?

- Line 96, Remove ‘e.g.’

- Line 236, “the maximum daily mean precipitation for each of the years (2018, 124.8 mm; 2019, 108.0 mm) were similar” How are these similar?

Reviewer #2: The manuscript entitled “Comparison of climatic factors on mosquito abundance at US Army Garrison Humphreys, Republic of Korea” analyzed the effect of climate conditions for different mosquito species. The study design and analysis are sound and appropriate. However, what kind of benefit can be delivered to vector control or disease prevention from the finding? The differences of maximum abundance temperature of each species are quite close.

Below are some comments:

Major issues:

1. The figure should demonstrate the landscape of different traps however, the white area was masked this information. It should be corrected. I understand it’s a military territory. If you are not allowed to show the landscape, at least describe the ecological feature for each trap.

2. The lag effects of temperature and precipitation on mosquito abundance have been discussed in many papers. Why do you decide to evaluate the effects from 7-13 days prior the collection only?

3. I suggest show the spatial distributions of mosquito abundance as a figure (map) for better presentation. The abundance of different species in the two years are significantly different.

4. Regarding the effect of ULV in Figure3, do you comparing mosquito abundances to one-day before the ULV applied? You have mentioned that mosquito samples were collected three-day a week, how do you compare the effect of ULV in a consecutive 4 days?

6. PLOS authors have the option to publish the peer review history of their article (what does this mean?). If published, this will include your full peer review and any attached files.

Reviewer #1: No

Reviewer #2: No

---

## [Author Response · Author response to Decision Letter 0]

18 Aug 2020

* We have included several figures and tables in the response to the reviewers. We would recommend the reviewers to open the attached Word file to read these figures and tables.

Reviewer #1

The authors collected a valuable dataset containing mosquito diversity and abundance information from an army base in the Republic of Korea (May-Oct 2018, May-Oct 2019). Mean daily temperature and daily precipitation data were obtained from a nearby weather station, and empirical mean temperature ranges for maximum abundance were determined for each mosquito species.

- The authors deeply appreciate a comprehensive review of the manuscript by the reviewer and for the valuable comments and suggestions. We have responded to the comments and the manuscript was revised accordingly.

Again, the dataset is large and unique, but I do have some concerns regarding the analysis:

Major comments

1. A time lag of 2 weeks was included in the model. However, do larval development times vary per species? And did you consider the potential age range of the adult population? A 2-week time lag assumes you only collect very young adults in the traps. How would the temperature-abundance relation look if different time lags were included (which ideally take into account the larval development times and mean/median age of the adult population for each species, if you can find literature on that)?

- We agree the reviewer’s comments. In general, the larval life cycle is known to be 10-14 days. Therefore, a lag of 2-week was applied based on the meteorological factors and in consideration of the adult period of young mosquitoes, which are generally known.

However, it also has been reported in a previous study that the most active period of mosquito adult activity was about 12 days (Burkett DA, Lee WJ, Lee KW, Kim HC, Lee HI, Lee JS, et al. Light, carbon dioxide, and octenol-baited mosquito trap and host-seeking activity evaluations for mosquitoes in a malarious area of the Republic of Korea. J Am Mosq Control Assoc. 2001;17:196-205).

In previous studies, they did not evaluate differences in the growth period of larvae for different mosquito species. In adult mosquitoes, the survival period of male adults is 2-3 weeks. For females, the oviposition cycle is about 3-4 days after a blood meal with overall survival of 4-5 weeks. Although mosquitoes were not classified by sex in this study, considering this period, the adult growth period was assumed to be the median period (average period) in this study (1-3 weeks for adult survival). The results of applying the temperature and cumulative precipitation for a lag of 3 to 5 weeks were compared (see Figures 1, 2, 3, below).

Figure 1. Relationship between the abundance of selected mosquito species and daily mean temperatures by generalized additive model (GAM), adjusted for the cumulative precipitation based on a 3-week lag effect during 2018 and 2019. The solid line is the estimated log (relative risk) as density of mosquitoes by species and dashed line is the 95% confidence interval. * Adjusted for a 3-week lag for cumulative precipitation, year, season, and application of insecticide by ULV fogging. RR, relative risk.

Figure 2. Relationship between the abundance of selected mosquito species and daily mean temperatures by generalized additive model (GAM), adjusted for the cumulative precipitation based on a 4-week lag effect during 2018 and 2019. The solid line is the estimated log (relative risk) as density of mosquitoes by species and dashed line is the 95% confidence interval. * Adjusted for a 4-week lag for cumulative precipitation, year, season, and application of insecticide by ULV fogging. RR, relative risk.

Figure 3. Relationship between the abundance of selected mosquito species and daily mean temperatures by generalized additive model (GAM), adjusted for the cumulative precipitation based on a 5-week lag effect during 2018 and 2019. The solid line is the estimated log (relative risk) as density of mosquitoes by species and dashed line is the 95% confidence interval. * Adjusted for a 5-week lag for cumulative precipitation, year, season, and application of insecticide by ULV fogging. RR, relative risk.

Compared with the results presented in the text, the range of the maximum abundance of mosquitoes by species after the adjustment of cumulative precipitation in the expanded lag was similar. However, as the range of lag expanded (lag 4, 5-weeks), the abundance of Anopheles Hyrcanus group and Culex pipiens showed a decrease as the temperature increased. Due to these effects, the relationship between the temperature of lag 4 and 5 weeks and total mosquito abundance was not statistically significant.

 In the previous results, when a lag of 2-weeks was applied, the relationship between relative mosquito abundance and temperature, adjusted for cumulative precipitation, was clearly shown. Therefore, we applied the 2-week lag for temperature. 

The strength of this study was the consecutive daily collection of mosquitoes conducted three days/week, so the numbers of collected mosquitoes, by species, were counted daily and not an average of 3-days. Most studies only collect mosquitoes once a week and analyzed data by matching the cumulative number of mosquitoes and meteorological indices per week. Therefore, we believe that our study has less limitation in deriving an accurate dose-response relationship between the temperature and the abundance of mosquitoes. 

2. On the topic of ‘time lags’: adult control applied with a ULV fogger was adjusted for all models, looking at a period of 1-4 days post-intervention. I am not familiar with the study area, but given the description it may be impossible to see an effect, given the amount of breeding sites (and thus mosquitoes) surrounding the army base. In addition, an effect is expected (i) after the next gonotrophic cycle has been completed, and not the next day, for many species, and (ii) 1-2 weeks later given the reduced numbers of eggs/larvae that go into the system. This makes me wonder if the current analysis is valid (to conclude ULV is not effective).

- The time lag was based on previous studies, e.g., Foley et al. (2012) who demonstrated a bi-modal distribution of Anopheles spp. in the ROK that corresponded with increased numbers of malaria positive mosquitoes. Masuoka et al. (2010) showed that as minimum temperatures increased in July, so did the populations of Culex tritaeniorhynchus. Additionally, Richards et al. (2010) showed a close relationship to rice paddies, which are adjacent to Camp Humphreys. However, these data were based on population numbers, not adult survival. 

 After taking the comments into account, the authors reanalyzed the effectiveness of the ULV fogger. Existing results analyzed the mosquito abundance applied to the delay effect (lag effect) of ULV fogging over the entire collection period, but this could be considered to be meaningless, according to the reviewer’s comment.

 For the reanalysis, we established two strategies. First, we compared the mosquito abundance of the day before and after the ULV fogging. Second, according to the reviewer’s comment, the ULV fogger was an anti-adult agent, but it was assumed that this could affect the larvae’s population. Therefore, we observed a decrease in mosquito abundance by applying a delay effect of 1 and 2 weeks.

 First, the abundance of mosquito on the day before and after the ULV fogging was compared by species (see Figure 4). Overall, the number of mosquitoes collected decreased on the following day compared to the day before the fogging and it was statistically significant (p<0.05). In the results classified by species, numbers of Aedes albopictus, Aedes lineatopennis, Culex pipiens, and Culex bitaeniorhynchus collected decreased significantly. For other species, the numbers of mosquitoes collected decreased the day after fogging, but the decrease in number was not statistically significant. 

 It is beneficial to understand that reductions in collected numbers depended on mosquito species, but that significant decreases in numbers collected were not observed for all species. The results for selected species are presented in Figure 4 of the manuscript.

To compare the residual effects of the day before and after ULV fogging, the numbers of mosquitoes collected at 1-week and 2-week were estimated by species. Overall, while the relative abundance of mosquitoes decreased after 1-week, the decrease was not statistically significant (see Table 1). Additionally, after 2-week, there was no residual effect of ULV fogging, but the numbers of mosquitoes collected increased (see Table 2). This indicates that ULV fogging has a minor temporary effect on reduction of adult mosquitoes.

Table 1. Comparison the mean numbers (range) of mosquitoes collected daily using Mosquito Magnet® Independence model® traps operated at 12 collection sites at USAG Humphreys 1 day before and 1 week after pesticide application using a ULV fogger during 2018.

Species Exposure Log (RR)* 95% CI p-value

Total Before 1 day ref 

 After 1 week -0.13 (-0.60 - 0.34) 0.63

Aedes albopictus Before 1 day ref 

 After 1 week 0.97 (0.56 - 1.39) 0.05

Aedes lineatopennis Before 1 day ref 

 After 1 week 0.49 (-0.36 - 1.35) 0.92

Aedes vexans nipponii Before 1 day ref 

 After 1 week 0.6 (-0.24 - 1.44) 0.30

Anopheles Hyrcanus Groupa Before 1 day ref 

 After 1 week 0.58 (-1.19 - 2.35) 0.59

Culex pipiens Before 1 day ref 

 After 1 week -0.47 (-1.26 - 0.30) 0.36

Culex orientalis Before 1 day ref 

 After 1 week 0.2 (-0.47 - 0.87) 0.64

Culex bitaeniorhynchus Before 1 day ref 

 After 1 week -0.04 (-1.23 - 1.15) 0.95

Culex tritaeniorhynchus Before 1 day ref 

 After 1 week -0.26 (-0.68 - 0.16) 0.34

Mansonia uniformis Before 1 day ref 

 After 1 week 1.04 (0.76 - 1.32) 0.04

*Adjusted for a lag of 2-weeks for daily mean temperature and cumulative precipitation, season, RR. Relative risk; CI, confidence interval. aAnopheles Hyrcanus Group includes An. sinensis s.s., An. belenrae, An. lesteri, An. kleini, and An. pullus that cannot be identified by morphological techniques.

Table 2. Comparison the mean numbers (range) of mosquitoes collected daily using Mosquito Magnet® Independence model® traps operated at 12 collection sites at USAG Humphreys 1 day before and 2 weeks after pesticide application using a ULV fogger during 2018.

Species Exposure Log (RR)* 95% CI p-value

Total Before 1 day ref 

 After 2-week 0.25 (-0.15 - 0.65) 0.44

Aedes albopictus Before 1 day ref 

 After 2-week 0.37 (-0.05 - 0.79) 0.69

Aedes lineatopennis Before 1 day ref 

 After 2-week 0.36 (-0.36 - 1.35) 0.88

Aedes vexans nipponii Before 1 day ref 

 After 2-week 2.88 (-0.24 - 1.44) 0.28

Anopheles Hyrcanus Groupa Before 1 day ref 

 After 2-week 1.41 (-1.19 - 2.35) 0.21

Culex pipiens Before 1 day ref 

 After 2-week 0.36 (-1.26 - 0.30) 0.87

Culex orientalis Before 1 day ref 

 After 2-week -0.13 (-0.47 - 0.87) 0.78

Culex bitaeniorhynchus Before 1 day ref 

 After 2-week 2.72 (-1.23 - 1.15) 0.19

Culex tritaeniorhynchus Before 1 day ref 

 After 2-week 0.58 (-0.68 - 0.16) 0.62

Mansonia uniformis Before 1 day ref 

 After 2-week 0.59 (0.76 - 1.32) 0.03

*Adjusted for a lag of 2-weeks for daily mean temperatures and cumulative precipitation, season, RR. Relative risk; CI, confidence interval. aAnopheles Hyrcanus Group includes An. sinensis s.s., An. belenrae, An. lesteri, An. kleini, and An. pullus that cannot be identified by morphological techniques.

3. Line 342 “Third, there is a limitation that classification by sex of mosquitoes was not possible according to the mosquito species, since the Mosquito Magnet® infrequently collects male mosquitoes”. I do not fully understand this sentence. Why was it not possible to distinguish the males from the females? Males have more fine hairs on their antennae, correct? This sentence came a bit ‘out of the blue’ for me, as I assumed you were showing results for the females, as they are relevant for disease transmission. How does this affect your results?

- This sentence was modified and the discussion that Mosquito Magnets infrequently collect males was deleted, since this did not pertain to the study. Males and females are easily identified, it is just that Mosquito Magnets collect mostly host-seeking mosquitoes and infrequently collects males, as occurs with New Jersey light traps.

4. In the discussion I am missing a section on climatic data (such as extremes, seasonal and daily fluctuations, micro-climatic conditions) and mosquito behaviors (thermal avoidance, resting biology).

- As a reviewer’s comment, we have put this in the discussion section.

Minor comments

1. Why the May-Oct periods? Is this a particular season?

- Mosquitoes are present as early as April at Camp Humphreys, but in very low numbers. Thus, adult surveillance begins the first week of May.

2. Line 96, Remove ‘e.g.’

- Following the reviewer’s comment, we deleted this word in line 96.

3. Line 236, “the maximum daily mean precipitation for each of the years (2018, 124.8 mm; 2019, 108.0 mm) were similar” How are these similar?

- While the duration and overall precipitation was much greater (4.1 X) during 2019, the daily maxims were similar, that is, the daily rainfall did not result in increased flooding caused by intense rainfall over a short period.

 

Reviewer #2

The manuscript entitled “Comparison of climatic factors on mosquito abundance at US Army Garrison Humphreys, Republic of Korea” analyzed the effect of climate conditions for different mosquito species. The study design and analysis are sound and appropriate. However, what kind of benefit can be delivered to vector control or disease prevention from the finding? The differences of maximum abundance temperature of each species are quite close.

- The authors deeply appreciate a comprehensive review of the manuscript by the reviewer and for the valuable comments and suggestions. The manuscript was revised accordingly.

Major comments

1. The figure should demonstrate the landscape of different traps however, the white area was masked this information. It should be corrected. I understand it’s a military territory. If you are not allowed to show the landscape, at least describe the ecological feature for each trap.

- We agree that the reviewer comments are important. However, this area is a military area of the United States, not the jurisdiction of the Republic of Korea, so a copy of the current map not supposed to be provided.

2. The lag effects of temperature and precipitation on mosquito abundance have been discussed in many papers. Why do you decide to evaluate the effects from 7-13 days prior the collection only?

- Reviewer’s comments are very important. In general, the larval life cycle is known as 10-14 days. Therefore, in this study, the meteorological factors for a lag of 2-weeks was applied in consideration of the adult period of young mosquitoes, which are generally known.

However, it has also been reported in the previous study that the most active period of mosquito adult activity is about 12 days (Burkett DA, Lee WJ, Lee KW, Kim HC, Lee HI, Lee JS, et al. Light, carbon dioxide, and octenol-baited mosquito trap and host-seeking activity evaluations for mosquitoes in a malarious area of the Republic of Korea. J Am Mosq Control Assoc. 2001;17:196-205).

Also, in previous studies, they did not evaluate differences in the growth period of larvae for different mosquito species. For adult mosquitoes, the survival period of male adults is 2-3 weeks. For females, the oviposition cycle is about 3-4 days after a blood meal and survival of 4-5 weeks. Although mosquitoes were not classified by sex in this study, considering this period, the adult growth period was assumed to be the median period (average period) in this study (1-3 weeks for adult survival). The results of applying temperature and cumulative precipitation of for a lag 3 to 5 weeks were compared (see Figures 1, 2, 3, above). For a lag of 4 and 5 weeks, see comments above. Based on previous results, a lag of 2-weeks was applied (see comment above). Strengths of the study are also discussed above.

3. I suggest show the spatial distributions of mosquito abundance as a figure (map) for better presentation. The abundance of different species in the two years are significantly different.

- We agree with the reviewer’s comment. In Figure 2 (in the manuscript), we mapped the abundance distribution by year as a proportion of the number of mosquito species/total number of mosquito populations collected. The proportion was mapped to 12 sites for selected species using the ArcGIS software. This was visualized to clearly show that the percentage collected by year is different for some species.

Figure 5. Distribution of the mean numbers of mosquitoes collected, by species, at USAG Humphreys, Pyeongtaek, Republic of Korea using Mosquito Magnet® Independence® model traps during 2018 and 2019. *Ratio was calculated as the number of mosquitoes collected by species/the number of total mosquitoes.

4. Regarding the effect of ULV in Figure3, do you comparing mosquito abundances to one-day before the ULV applied? You have mentioned that mosquito samples were collected three-day a week, how do you compare the effect of ULV in a consecutive 4 days?

- After taking the comments into account, the authors reanalyzed the effectiveness of the ULV fogger (See comments and figures above). 

6. PLOS authors have the option to publish the peer review history of their article (what does this mean?). If published, this will include your full peer review and any attached files.

Do you want your identity to be public for this peer review? For information about this choice, including consent withdrawal, please see our Privacy Policy.

Reviewer #1: No

Reviewer #2: No

---

## [Decision Letter · Decision Letter 1]

25 Sep 2020

Comparison of climatic factors on mosquito abundance at US Army Garrison Humphreys, Republic of Korea

PONE-D-20-17361R1

Dear Dr. Cheong,

We’re pleased to inform you that your manuscript has been judged scientifically suitable for publication and will be formally accepted for publication once it meets all outstanding technical requirements.

Kind regards,

Silvie Huijben

Academic Editor

PLOS ONE

Additional Editor Comments (optional):

Reviewers' comments:

Reviewer's Responses to Questions

**Comments to the Author**

1. If the authors have adequately addressed your comments raised in a previous round of review and you feel that this manuscript is now acceptable for publication, you may indicate that here to bypass the “Comments to the Author” section, enter your conflict of interest statement in the “Confidential to Editor” section, and submit your "Accept" recommendation.

Reviewer #2: All comments have been addressed

2. Is the manuscript technically sound, and do the data support the conclusions?

Reviewer #2: Yes

3. Has the statistical analysis been performed appropriately and rigorously? 

Reviewer #2: Yes

4. Have the authors made all data underlying the findings in their manuscript fully available?

Reviewer #2: Yes

5. Is the manuscript presented in an intelligible fashion and written in standard English?

Reviewer #2: Yes

6. Review Comments to the Author

Reviewer #2: The authors have addressed all the comments. The quality of manuscript has been improved.

7. PLOS authors have the option to publish the peer review history of their article (what does this mean?). If published, this will include your full peer review and any attached files.

Reviewer #2: No

---

## [Editor Report · Acceptance letter]

9 Oct 2020

PONE-D-20-17361R1 

Comparison of climatic factors on mosquito abundance at US Army Garrison Humphreys, Republic of Korea 

Dear Dr. Cheong:

I'm pleased to inform you that your manuscript has been deemed suitable for publication in PLOS ONE. Congratulations! Your manuscript is now with our production department. 

Kind regards, 

on behalf of

Dr. Silvie Huijben 

Academic Editor

PLOS ONE